# Clinical presentation and outcomes of pre-eclampsia and eclampsia at a national hospital, Kenya: A retrospective cohort study

Charity Ndwiga[1]*, George Odwe[1], Sripad Pooja[2], Omondi Ogutu[3], Alfred Osoti[3], Charlotte E. Warren[2]

1 Population Council, Nairobi, Kenya, 2 Population Council, Washington, DC, United States of America, 3 OBGyn Department, University of Nairobi, Nairobi, Kenya

* cndwiga@popcouncil.org

## Abstract

### Background

Hypertensive disorders in pregnancy including pre-eclampsia are associated with maternal and newborn mortality and morbidity. Early detection is vital for effective treatment and management of pre-eclampsia. This study examines and compares the clinical presentation and outcomes between early- and late-onset pre-eclampsia over a two year period.

### Methods

A retrospective cohort study design which examines socio-demographic characteristics, treatment, outcomes, and fetal and maternal complications among women with early onset of pre-eclampsia (EO-PE) and late onset of pre-eclampsia (LO-PE). De-identified records of women who attended antenatal, intrapartum and postnatal care services and experienced pre-eclampsia at Kenyatta National teaching and referral hospital were reviewed. We used chi square, t-test, and calculated odds ratio to determine any significant differences between the EO-PE and LO-PE cohorts.

### Results

Out of 620 pre-eclamptic and eclamptic patients' records analyzed; 44 percent (n = 273) exhibited EO-PE, while 56 percent had late onset. Women with EO-PE compared to LO-PE had greater odds of adverse maternal and perinatal outcomes including hemolysis elevated liver enzymes and low platelets (HELLP) syndrome (OR: 4.3; CI 2.0–10.2; p<0.001), renal dysfunction (OR; 1.7; CI 0.7–4.1; p = 0.192), stillbirth (OR = 4.9; CI 3.1–8.1; p<0.001), and neonatal death (OR: 8.5; CI 3.8–21.3; p<0.001). EO-PE was also associated with higher odds of prolonged maternal hospitalization, beyond seven days (OR = 5.8; CI 3.9–8.4; p<0.001), and antepartum hemorrhage (OR = 5.8; CI 1.1–56.4; p<0.001). Neonates born after early onset of pre-eclampsia had increased odds of respiratory distress (OR = 17.0; CI 9.0–32.3, p<0.001) and birth asphyxia (OR: 1.9; CI 0.7–4.8; p = 0.142).

**Data Availability Statement:** There are institutional legal restrictions on sharing de-identified data sets, however data is available on request. Request may be sent to Population Council, Dataverse, email;

publications@popcouncil.org for information on data access.

**Funding:** The study is made possible by the generous support of the American people through the United States Agency for International Development (USAID) under the terms of USAID APS-OAA-14-00048. The contents of this manuscript are the sole responsibility of the authors, Ending Eclampsia project and Population Council and do not necessarily reflect the views of USAID or the United States Government.

**Competing interests:** The authors declare no competing interests.

**Abbreviations:** ANC, Antenatal Care; dBP, Diastolic Blood pressure; EO-PE, Early Onset Pre-Eclampsia; HDP, Hypertensive Disorders of Pregnancy; KNH, Kenyatta National Hospital; LMIC, Low- and Middle-Income Country; LO-PE, Late Onset Pre-Eclampsia; MgSO$_4$, Magnesium Sulphate; NCU, Newborn Care Unit; NICU, Neonatal Intensive Care Unit; Ob/Gyn, Obstetricians and Gynecologists; OR, Odds Ratio; PNC, Postnatal Care; sBP, Dystolic Blood pressure; WHO, World Health Organization.

## Conclusions

The profiles and outcomes of women with EO-PE (compared to late onset) suggest that seriousness of morbidity increases with earlier onset. To reduce adverse neonatal and maternal outcomes, it is critical to identify, manage, referral and closely follow-up pregnant women with pre-eclampsia throughout the pregnancy continuum.

## Ethical approval

This study protocol was approved by Population Council's research ethics Institutional Review Board, Protocol 813, and KNH-UoN Ethics and Research Committee, Protocol 293/06/2017.

## Background

Globally, about five to ten percent of women experience hypertensive disorders in pregnancy (HDPs) [1], making it the second leading cause of maternal mortality and morbidity. HDPs include pre-eclampsia and eclampsia, gestational hypertension, and chronic hypertension. Pre-eclampsia, that is—hypertension with proteinuria after 20 weeks' pregnancy gestation, complicates about five percent of pregnancies [2]. Majority of the estimated 70,000 to 80,000 annual maternal, and 500,000 annual perinatal pre-eclampsia-related deaths occur in low and middle-income countries (LMICs) [3,4]. In Kenya, the 2014 confidential inquiry into maternal death revealed HDP as the third leading cause of maternal mortality, accounting for 20 percent of maternal deaths [5]. The prevalence of pre-eclampsia in Kenya is estimated to range between 5.6 to 6.5 percent [6,7], though proportions are likely higher in rural areas. While HDPs including PE are recognized as a leading cause of maternal and newborn mortality and morbidity, there is an increasing emphasis on early detection and management. Uncertainty around the differences between women who develop PE before and after 34 weeks gestation remains underexplored in low- and middle-income settings.

Pre-eclampsia and eclampsia have a devastating effect on pregnant women, their fetuses, and newborns. Women with pre-eclampsia suffer severe morbidity and mortality due to placental abruption, pulmonary edema, acute renal failure, and other organ damage [8,9]. Moreover, newborns of women with pre-eclampsia have approximately twice the risk of neonatal death, and increased risks of low Apgar scores, seizures, neonatal encephalopathy, and neonatal intensive care admission [10,11]. Women with pre-eclampsia who deliver before 37 weeks of pregnancy are also likely to have babies with low birth weight [12,13]. Recent studies in sub-Saharan Africa identified risk factors for pre-eclampsia including: primiparity and multiple parity (more than 4), low plasma vitamin C, low levels of vitamin D, chronic hypertension, family history of hypertension, as well as low levels of education [14,15].

Early detection and management of pre-eclampsia is essential, but few studies have examined prevalence or incidence of early versus late onset of pre-eclampsia in LMICs, including Kenya. Late detection and poor management of pre-eclampsia in primary healthcare facilities negatively affect newborn and maternal health outcomes. Understanding the pre-eclampsia burden is necessary for improving care for women experiencing both early and late onset pre-eclampsia and eclampsia. We therefore examined Kenya's pre-eclampsia and eclampsia clinical spectrum, including presentation at admission, management of complications, and maternal and newborn health outcomes at the maternity unit of a national referral hospital. The

purpose of this study is to describe the profile of women with pre-eclampsia and eclampsia and examine the differences in maternal and perinatal characteristics and health outcomes between early onset pre-eclampsia (EO-PE: less than 34 weeks gestation) and late onset pre-eclampsia, (LO-PE: greater than or equal to 34 weeks gestation)—for women who received antenatal; care (ANC), intrapartum care, or postnatal care (PNC) at Kenyatta National Hospital (KNH).

## Analytic framework

EO-PE is associated with higher morbidities and more severe maternal and neonatal/fetal outcomes than LO-PE [16,17]. To reduce poor maternal and perinatal outcomes, including prematurity, prompt and effective care, early delivery for women with early onset disease is paramount [18,19]. We used an analytic framework, based on existing evidence and clinical practice, (Fig 1) to guide the retrospective review of in-patient records of women who experienced both EO-PE and LO- PE in a Kenyan context.

## Methods

### Study design

This study adopts a retrospective cohort design that examined socio-demographic characteristics, treatment, outcomes, and fetal and maternal complications among women with EO-PE and LO-PE. It draws on a review of de-identified records of women who attended ANC, intrapartum care, and PNC services and experienced pre-eclampsia, over a two-year period (September 2015 to October 2017).

### Study site

This study was conducted at KNH, the largest teaching and referral hospital in Kenya and East Africa with a capacity of 1,800 beds. KNH also serves as a primary hospital for Nairobi County residents in addition to referred patients from different parts of Kenya. The labor ward operates 24 hours a day and conducts about 1,255 deliveries per month while ANC clinic services about 2,164 clients per month. Pregnant and postnatal women with complications, including medical or surgical complications, receive multidisciplinary care either within the maternity unit or in other units. The maternity unit's newborn care unit (NBU) has 60 beds/incubators, and the neonatal intensive care unit (NICU) has five incubators/respirators. About 100 babies are admitted per day to KNH's NBU and stay for an average of 14 days.

### Study population

This study population comprises women who received ANC, intrapartum care, or PNC (up to 12 weeks following birth) at KNH or those who were referred to KNH preceding, during or following delivery, and suffered from pre-eclampsia and eclampsia after 20 weeks' gestation. Eligibility criteria included PE/E diagnosis with high blood pressure, proteinuria, or evidence of end organ damage. Women who had hypertension but not pre-eclampsia were excluded from the study.

### Sample size and sampling

A two-proportion sample size calculation, to detect 30 percent difference with 95 percent confidence and 80 percent power [20], accounting for missing cases, resulted in a sample of 620 women's records for review. The calculation was based on the main outcome: the incidence of adverse events after EO-PE and LO-PE among pregnant or postnatal women presenting for

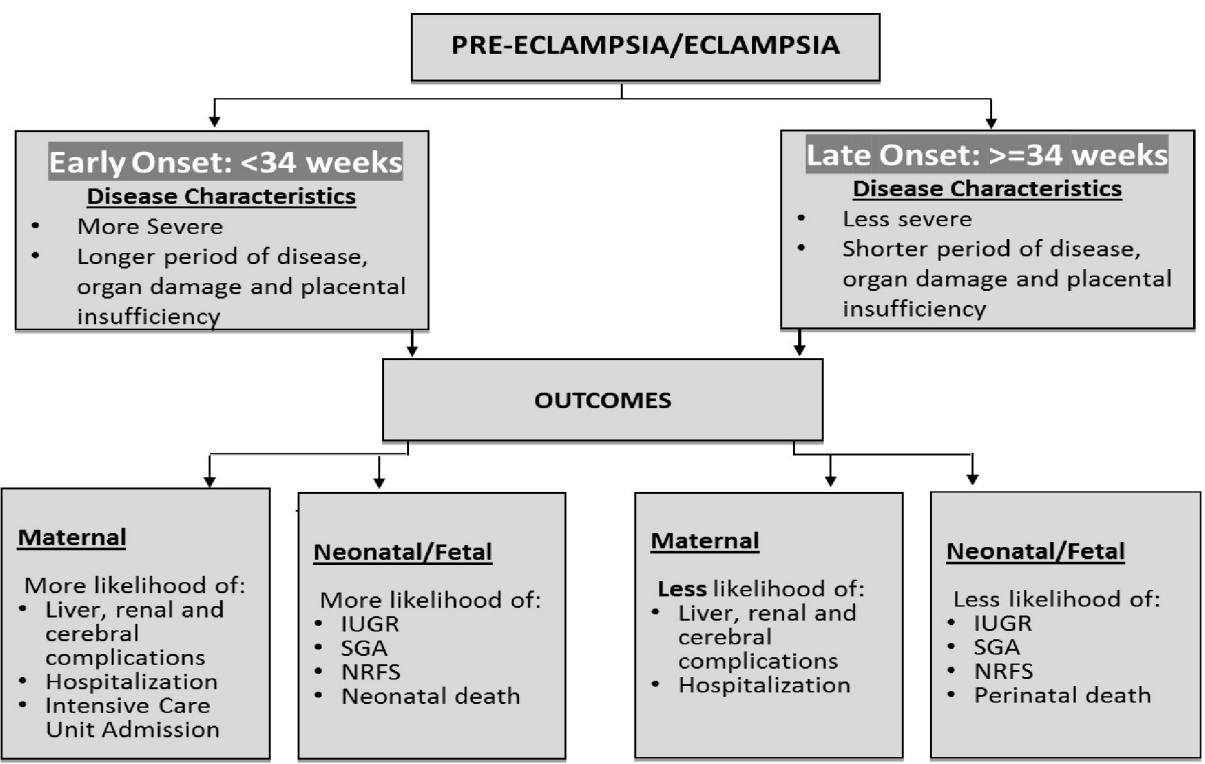

IUGR, Intrauterine growth restriction; SGA, Small-for-Gestational-Age; NRFS, Non-reassuring Fetal Status

**Fig 1. Analytic framework.**

care at KNH for the indexed pregnancy. Prior studies in Kenya revealed EO-PE and LO-PE incidence rates of 0.38 percent and 2.7 percent [6,7], respectively; considering our high risk setting in a referral institution, we used incidence estimates of 0.4 percent EO-PE and four percent LO-PE to estimate our sample.

We obtained a list of pre-eclampsia and eclampsia cases in ANC, delivery and PNC units. Of 1,529 eligible in-patient records identified, 191 (12.5%) women experienced eclampsia and the remainder experienced pre-eclampsia. All eclampsia cases were included since the number was small. We then randomly sampled the inpatient records describing pre-eclampsia, stratified by month and year of observation resulting to 909 (59.5%). After exclusion of a total of 480 (43.6%) case files (due to missing or incomplete data), we achieved a sample of 620 PE/E case files for analysis. Fig 2 shows the study's sampling method and record review procedure for the study period.

## Data extraction and analysis

Data were extracted from de-identified maternity inpatient records focusing on the time of diagnosis of PE/E and the time of any treatment or intervention that enabled measurement of clinical management and care. Extractions assessed eligibility (EO-PE or LO-PE) by consensus review of the in-patient records and included: medical records, clinician notes or antenatal cards for any exposure in the month preceding delivery, gestational age by date, and ultrasound results. Data on maternal outcomes included eclampsia, cerebral vascular accident, other organ damage, dialysis, prolonged hospitalization admission including high dependency

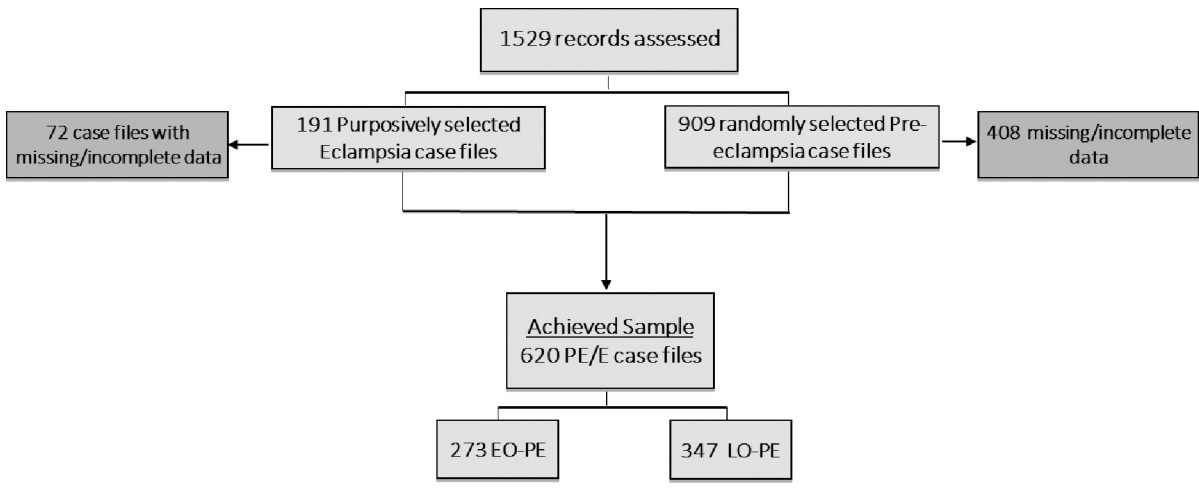

**Fig 2. Flow diagram of patient's records sample selection.**

or intensive care unit, and mortality. Perinatal outcomes included low birth weight, prematurity, respiratory distress syndrome, jaundice, admission to NCU and/or NICU, and mortality. Pre-term births include all live births prior to 37 weeks' gestation. Extremely pre-term babies occur prior to 28 weeks, very pre-term between 28 and 32 weeks, and moderate to late pre-term between 32 and 37 weeks [18,19].

Descriptive statistics (frequency and percentage) and measures of central tendency (mean, standard deviations, median) are reported. We used chi square, t-test, and calculated odds ratios to determine any significant differences between the EO-PE and LO-PE cohorts. Descriptive statistics and bivariate analysis compared between EO-PE and LO-PE for maternal outcomes and neonatal outcomes respectively using Stata® 15.

## Results

### Background characteristics

A total of 620 records were reviewed. Forty-four (44%) of the records (n = 273; *45 had eclampsia*) exhibited EO-PE, while 56 percent (n = 347; *74 had eclampsia*) had LO-PE. Among women presenting with EO-PE, 124 (45.4%) were less than 30 weeks pregnant, and 149 (54.6%) were between 30 to 33 weeks. For women presenting with LO-PE, 39 percent were of 34 to 36 weeks' gestation, while 61 percent were more than 36 weeks. There were no statistically significant differences in the socio-demographic characteristics (parity, marital status, education, work status, and place of residence) between the EO-PE and LO-PE groups. A total of 126 (20.3%) women with pre-eclampsia and eclampsia resided in other counties while the rest lived in Nairobi county (Table 1).

### Obstetric risk factors

Women with EO-PE were more likely to have chronic hypertension or family history of hypertension compared to those with LO-PE (Fig 3). Prior experience of PE was also a major risk factor (1 in 10) for both EO-PE and LO-PE. Overall, only a few women had gestational diabetes mellitus, pre-gestational diabetes mellitus and prior eclampsia (< 2%).

**Table 1. Background characteristics of study participants.**

| Characteristics | Early Onset (PE) (N = 273) | | Late Onset (PE) (N = 347) | | Total (N = 620) | | P-value |
|---|---|---|---|---|---|---|---|
| Age (years) | | | | | | | |
| Mean [SD] | 29.0 | [6.1] | 28.0 | [6.4] | 28.0 | [6.3] | 0.008 |
| | n | % | n | % | n | % | |
| Parity | | | | | | | |
| 0 | 30 | 11.0 | 42 | 12.1 | 72 | 11.6 | 0.060 |
| 1 | 88 | 32.2 | 143 | 41.2 | 231 | 37.3 | |
| 2 | 78 | 28.6 | 73 | 21.0 | 151 | 24.4 | |
| 3+ | 77 | 28.2 | 89 | 25.6 | 166 | 26.8 | |
| Gravidity | | | | | | | |
| 1 | 72 | 26.4 | 150 | 43.2 | 222 | 35.8 | **<0.001** |
| 2 | 80 | 29.3 | 83 | 23.9 | 163 | 26.3 | |
| 3 | 66 | 24.2 | 59 | 17.0 | 125 | 20.2 | |
| 4+ | 55 | 20.1 | 55 | 15.9 | 110 | 17.7 | |
| Formal Education | | | | | | | |
| < = Primary^ | 82 | 30.0 | 110 | 31.7 | 192 | 31.0 | 0.700 |
| Secondary | 112 | 41.0 | 133 | 38.3 | 245 | 39.5 | |
| Tertiary | 79 | 28.9 | 104 | 30.0 | 183 | 29.5 | |
| Employment | | | | | | | |
| Employed† | 134 | 49.1 | 161 | 46.4 | 295 | 47.6 | 0.500 |
| Unemployed‡ | 139 | 50.9 | 186 | 53.6 | 325 | 52.4 | |
| Marital Status | | | | | | | |
| Married | 220 | 80.6 | 291 | 83.9 | 511 | 82.4 | 0.300 |
| Single | 53 | 19.4 | 56 | 16.1 | 109 | 17.6 | |
| Place of Residence | | | | | | | |
| Nairobi County | 205 | 75.1 | 289 | 83.3 | 494 | 79.7 | **0.010** |
| Others | 68 | 24.9 | 58 | 16.7 | 126 | 20.3 | |
| **Gestational age at onset, weeks** | | | | | | | |
| | **273** | **SD** | **347** | **SD** | **620** | **SD** | **<0.001** |
| Mean (SD) | 29.2 | 3.2 | 37.3 | 2.1 | 33.7 | 4.8 | |

Significant results (P<0.05) in bold color

* (n/N*100)

^None and primary education

† Includes Business, self-employed

‡ includes housewife

**includes spontaneous vaginal delivery, assisted delivery and breech delivery

## Pre-eclampsia characteristics

Women with EO-PE were more likely to experience pre-eclampsia with severe complications than women with LO-PE. Eclampsia incidence was not statistically significantly associated with the onset of pre-eclampsia. However, women with EO-PE were more likely to experience severe pre-eclampsia and show highest systolic blood pressure (sBP), that is, sBP greater than 180 mmHg, than women with LO-PE. On the other hand, women with LO-PE were more likely than those with EO-PE to experience non-severe pre-eclampsia, have highest SGOT (>70) and highest Creatine (>100) and show lowest Platelets (<100000), higher diastolic blood pressure (dBP), and lower creatinine. Results show no significant differences among women with low hemoglobin (i.e. hemoglobin level <10) between EO-PE and LO-PE (Table 2)

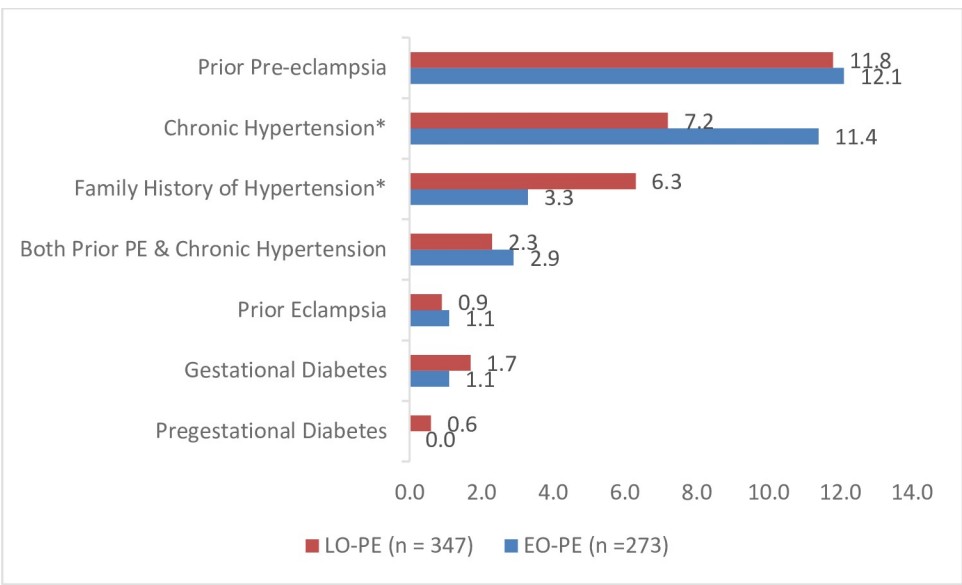

**Fig 3. Proportion of women experiencing obstetric risk factors by onset of pre-eclampsia at KNH, September 2015 to October 2017.**

## Pregnancy and fetal outcomes

As expected, mean gestational age at delivery was lower for women with EO-PE than women with LO-PE. Live births and cesarean sections were more frequent among women with LO-PE than those with EO-PE. Poor fetal outcomes were more frequent among women with EO-PE including deaths, stillbirths, low birth weight and admission to the NBU (Table 3)

## Neonatal and maternal complications

Neonates born to women with EO-PE were more likely than those born to women with LO-PE to experience the following complications: neonatal jaundice, necrotizing enterocolitis, neonatal convulsions, birth asphyxia, and respiratory distress syndrome, and requiring breathing support through continuous positive airway pressure (Table 3). In terms of maternal complications, women with EO-PE were more likely to experience antepartum hemorrhage, and

**Table 2. Proportion of women by severity of pre-eclampsia and eclampsia and blood results by onset of disease.**

| Severity of disease and blood results | EO-PE (n = 273) | LO-PE (n = 347) | OR [95% CI] | p-value |
|---|---|---|---|---|
| | n (%) | n (%) | | |
| Pre-Eclampsia | 11 (4.2) | 63 (18.1) | 0.2(0.1–0.4) | <0.001 |
| Severe Pre-Eclampsia | 217 (79.3) | 210 (60.5) | 2.5(1.7–3.7) | <0.001 |
| Eclampsia | 45 (16.5) | 74 (21.5) | 0.7(0.5–1.1) | 0.160 |
| Highest sBP >180 mmHG | 128 (46.7) | 113 (32.6) | 1.9(1.3–2.6) | <0.001 |
| Lowest dBP <90 mmHG | 67 (24.5) | 145 (44.4) | 0.4(0.3–0.6) | <0.001 |
| Highest SGOT (>70) | 77 (28.2) | 130 (37.5) | 0.7(0.5–0.9) | 0.020 |
| Highest Creatine (>100) | 160 (58.6) | 274 (79.9) | 0.4(0.3–0.5) | <0.001 |
| Lowest Hemoglobin levels (<10) | 18 (6.7) | 30 (9.5) | 0.7(0.4–1.4) | 0.340 |
| Lowest Platelets (<100000) | 68 (24.9) | 110 (31.7) | 0.7(0.5–1.0) | 0.060 |

SGOT—serum glutamic-oxaloacetic transaminase; OR—Odds Ratios; CI—Confidence Interval

**Table 3. Pregnancy and fetal outcomes among women presenting with EO-PE and LO-PE.**

| Foetal and maternal outcomes | EO-PE (n = 242) | LO-PE (n = 341) | OR [95% CI] | p-value |
|---|---|---|---|---|
| **Mean gestational age at delivery in weeks** (SD) | 30.5 (3.6) | 37.4 (2.1) | | <0.001 |
| **Mode of delivery** * | | | | |
| Spontaneous vaginal delivery | 94(38.8) | 86(25.2) | 1.6(1.1–2.3) | 0.009 |
| Assisted vaginal delivery | 2(0.8) | 0(0.0) | N/A | N/A |
| Caesarean section | 145(59.9) | 254(74.5) | 0.4(0.3–0.6) | <0.001 |
| Breech delivery | 1(0.4) | 1(0.3) | N/A | N/A |
| **Pregnancy outcomes** | | | | |
| Birthweight (<2.5kg) | 224(92.6) | 115(33.7) | 2.6(1.3–5.3) | 0.003 |
| Neonatal deaths | 41(16.9) | 8(2.3) | 8.5(3.8–21.3) | 0.002 |
| Stillbirth | 82(33.9) | 32(9.4) | 4.9(3.1–8.0) | <0.001 |
| Live birth | 151(62.4) | 310(90.6) | 0.2(0.1–0.3) | <0.001 |
| **NBU admissions** | 133(55.0) | 83(24.3) | 3.8(2.6–5.5) | <0.001 |
| **NICU admissions** | 2(0.2) | 2(0.6) | N/A | N/A |

*37 women (31 EO-PE AND 6 LO-PE) were discharged from ANC; N/A-results not available due to few cases

HELLP syndrome (hemolysis, elevated liver enzymes, and low platelet count) and stayed in hospital longer due to complications compared to women with LO-PE. The proportion of women with pre-eclampsia who experienced any complication was statistically significantly higher among EO-PE group than LO-PE group. Six women died (1 with EO-PE, and 5 with LO-PE; all six were admitted to KNH in an eclamptic state, Table 4).

## Medicines used in the management of pre-eclampsia and eclampsia

**Prophylaxis.** Few women were prescribed prophylactic aspirin which is recommended from 12 weeks' gestation for women at high risk of developing pre-eclampsia. Only three percent of women received calcium supplement during pregnancy—which is recommended for women with a low calcium diet, to prevent complications associated with pre-eclampsia.

**Treatment of hypertension.** Most women with both EO-PE and LO-PE received treatment for hypertension with oral Alpha Methyldopa (Aldomet) and Nifedipine, throughout the pregnancy continuum (ANC, labor and delivery, PNC). Alpha Methyldopa is on the WHO essential medicines list for use in pregnant women. Over 90% of women prescribed and using Nifedipine after delivery had used Methyldopa during pregnancy and labor (results not shown). If Alpha Methyldopa and Nifedipine are unavailable, intravenous Hydralazine is given as second line drug.

**Management of pre-eclampsia.** Magnesium sulphate ($MgSO_4$) is recommended as the medicine of choice by WHO for preventing and managing seizures in severe pre-eclampsia and eclampsia [21]. More than half of women's case notes reviewed showed that $MgSO_4$ was prescribed during the antenatal period (EO-PE 60%, LO-PE 51%). However, the continued use of Diazepam and Phenytoin to manage eclamptic seizures also persist (3.2% during pregnancy, 1.4% during labor and 3.9% postnatally). Three of the women who died had received a pre-referral dose of $MgSO_4$. Table 5 shows the use of antihypertensive drugs, MgSO4, and other drugs recorded during pregnancy, labor and delivery, and the postnatal period.

## Discussion

This study contributes to our understanding of early verses late onset pre-eclampsia, including the management and outcomes of these conditions in a LMIC context. In reviewing in-patient

**Table 4. Neonatal and maternal complications.**

| Complications | EO-PE (n = 96) | LO-PE (n-289) | OR (95% CI) |
|---|---|---|---|
| | n (%) | n (%) | |
| **Proportion of neonates with complications*** | **96(39.8)** | **56(16.4)** | 2.4(1.6–3.5) |
| **Neonatal complications** | | | |
| Respiratory distress syndrome (RDS)* | 56(58.3) | 22(7.6) | 16.9(9.0–32.2) |
| Neonatal jaundice† | 22(22.9) | 15(5.2) | 5.4(2.5–11.8) |
| Asphyxia | 9(9.4) | 15(5.2) | 1.9(0.7–4.8) |
| Neonatal convulsion | 3(3.1) | 4(1.4) | N/A |
| Necrotizing enterocolitis (NEC) | 6(6.3) | 0(0.0) | N/A |
| **Maternal complications** | **(n = 232)** | **(n = 341)** | |
| Antepartum Hemorrhage-† | 8 (3.3) | 2 (0.6) | 5.8(1.1–56.8) |
| HELLP* | 29 (13.0) | 10 (3.0) | 4.3(2.0–10.1) |
| Postpartum Hemorrhage | 9 (3.7) | 15 (4.4) | 0.8(0.3–2.1) |
| Renal Failure | 14 (5.8) | 12 (3.5) | 1.7(0.7–4.1) |
| Cerebral Vascular Accident /Stroke | 4 (1.7) | 3 (0.9) | N/A |
| Retinal Detachment/Blindness | 0 (0.0) | 0 (0.0) | N/A |
| Disseminated Intravascular Coagulation (DIC) | 0 (0.0) | 2 (0.6) | N/A |
| Pulmonary Oedema | 0 (0.0) | 1 (0.3) | N/A |
| Deep Vein Thrombosis | 0 (0.0) | 2 (0.6) | N/A |
| Dialysis required | 8 (3.0) | 6 (2.0) | 1.9(0.6–6.6) |
| Transfusion required | 24 (9.9) | 23 (6.7) | 1.5(0.8–2.9) |
| Death | 1 (0.4) | 5 (2.0) | N/A |
| **Proportion of women with PE who experienced any complication*** | 164(60.1) | 104(29.9) | 3.5(2.5–5.0) |
| **Length of stay** | | | |
| Intensive Care Unit/ High Dependency Unit admission | 8 (3.0) | 10 (3.0) | 1.13(0.4–3.2) |
| Prolonged Hospital Stay > 1 week* | 155 (67.0) | 81 (24.0) | 5.7(3.9–8.4) |

* p <0.001

† p<0.01; N/A–not available due to fewer/no cases; OR—Odds Ratios; CI—Confidence Interval. Analysis excludes 37 women discharged at ANC

records of women at a national referral hospital in Kenya, notable distinctions in the clinical presentation and management between EO-PE and LO-PE elucidates the complexity in trying to reduce adverse maternal and fetal outcomes in a low-resource setting. As demonstrated elsewhere [18,22], timely referral is critical in reducing poor maternal and fetal outcomes and show that if delayed, care is compromised even in a relatively well-resourced tertiary facility with clinical expertise and available equipment.

Obstetric risk factors such as prior experience of pre-eclampsia, eclampsia and chronic hypertension may elevate a woman's risk for EO-PE. Similar to other studies [23–25], Our findings show that while more women who experience LO-PE had gestational diabetes and a family history of hypertension, women with EO-PE had higher rates of chronic hypertension. Severe pre-eclampsia, high sBP (>180), and high creatinine levels were all associated with EO-PE as reported elsewhere [26,27].

A high proportion of pregnancies (more than two thirds) resulted in cesarean section—with significantly more women with LO-PE (75%) than with EO-PE (60%), although there were no significant differences in perinatal outcomes by mode of delivery. Evidence from other studies comparing outcomes for infants born to EO-PE women, through vaginal or c-section delivery, is weak [28] creating challenges for obstetricians when determining the optimal mode of delivery for women with pre-eclampsia or eclampsia [3,29,30]. Babies born to

**Table 5. Management of pre-eclampsia and eclampsia across pregnancy continuum.**

| Medicine | Pregnancy | | Labor and Delivery* | | Postnatal* | |
|---|---|---|---|---|---|---|
| | EO-PE (n = 273) n (%) | LO-PE (n = 347) n (%) | EO-PE (n = 242) n (%) | LO-PE (n = 341) n (%) | EO-PE (n = 242) n (%) | LO-PE (n = 341) n (%) |
| *Antihypertensive medicines* | | | | | | |
| Methyldopa (oral) | 259(94.9) | 289(83.3) | 230(95.0) | 287(84.2) | 230(95.0) | 295(86.5) |
| Nifedipine (oral) | 257(94.1) | 276(79.5) | 227(93.8) | 279(81.8) | 223(92.2) | 293(85.9) |
| Labetalol (oral) | 10(3.7) | 6(1.7) | 4(1.7) | 0.0 | 4(1.7) | 0.0 |
| Hydralazine (IV) | 69(25.3) | 33(9.5) | 34(14.1) | 18(5.3) | 21(8.7) | 15(4.4) |
| *Anticonvulsants* | | | | | | |
| MgSO4 | 164(60.1) | 178(51.3) | 99(40.9) | 133(39.0) | 49(20.3) | 89(26.1) |
| Diazepam | 3(1.1) | 0.0 | 0.0 | 2(0.6) | 1(0.4) | 2(0.6) |
| Phenytoin | 5(1.8) | 1(0.3) | 2(0.8) | 2(0.6) | 5(2.1) | 6(1.8) |
| Lasix (diuretic) | 10(3.7) | 3(0.9) | 6(2.5) | 4(1.2) | 13(5.4) | 4(1.2) |
| *Prophylaxis* | | | | | | |
| Junior Aspirin | 15(5.5) | 3(0.9) | 1(0.4) | 0.0 | 0.0 | 0.0 |
| Calcium | | | | | | |

IV, Intravenous.

*Analysis excludes 37 women discharged at ANC

women with EO-PE were more likely to experience respiratory distress syndrome, necrotizing enterocolitis, and neonatal jaundice. This resonates with other studies that shows impaired fetal growth along with higher perinatal mortality and morbidity among women with EO-PE than among non-PE or LO-PE women [23,31]. Previous research similarly shows higher rates of premature births, NICU admission, severe neonatal morbidity and neonatal deaths among newborns of women with EO-PE [32–34].

## Programming that affects prevention pre-eclampsia and eclampsia

Prevention, early detection and management of pre-eclampsia and eclampsia is critical throughout the continuum of care [21]. Effective preconception care and early and regular ANC provide platforms to screen for pre-existing conditions such high blood pressure and educate soon-to-be-pregnant and pregnant woman on pre-eclampsia risk factors [27,29]. In addition to self-referrals by well-informed women, better diagnostic and surveillance capacities are necessary. ANC also involves specialized follow-up and control of chronic and gestational hypertension, availability and correct administration of drugs, timely delivery with optimum neonatal care, and critical and intensive maternal care [35–37]. Adherence to stricter treatment protocols for controlling hypertension in pregnancy and appropriate use of recommended drugs regimens for women exhibiting blood pressure of 150/100, has been shown to reduce pregnancy risks especially in EO-PE [38–40].

More women with EO-PE experienced maternal complications such as HELLP syndrome and antepartum hemorrhage than LO-PE women. Sub-optimal management, particularly at earlier stages of pregnancy, can lead to death [41]. Moreover, delays in seeking care from home and late referral from lower level facilities to the tertiary level also contributes to poor maternal outcomes [42]. Around one fifth of women diagnosed with PE/E (20%) were referred to KNH from outside Nairobi County, most from neighboring counties. A study in Tanzania showed a higher risk of maternal mortality among eclamptic women when they presented in a critically ill or moribund state [43]. In other studies, most women received at a referral hospital

with eclampsia had not received any ANC or had been poorly managed in lower level facilities, with poor fetal and maternal outcomes [28,30,31,44,45]. These results lend credence to the importance of early referral for women with pre-eclampsia and eclampsia. A health system approach, focused on skilled provider availability, specifically midwives, with training in emergency obstetric and neonatal care, in addition to facility preparedness for measuring blood pressure, effective blood pressure management, and close follow up throughout pregnancy, with timely referrals, can contribute to reductions in maternal (and newborn) deaths [46–48].

Contrary to other studies, our study shows similar use of antihypertensives and $MgSO_4$ among the two cohorts (e.g. EO-PE and LO-PE) throughout pregnancy, labor and delivery, and postnatal period. Discontinuity in HDP management often occurs in the postnatal period, highlighting the need to build on studies reporting incidences of postnatal hypertension [17,49]. Satisfactory PNC includes careful clinical evaluation of blood pressure, among other postnatal vital signs, during the hospital stay, with delayed discharge if hypertension continues. In addition to women's awareness of danger signs, such as headache and dizziness, and continued use of antihypertensives after they return home [21,50–52]. Our study shows that over 90% of women that had used Alpha Methyldopa in pregnancy and labor used Nifedipine in the postnatal period. Despite this, 80% of women appeared to continue to be prescribed Alpha Methyldopa in the postnatal period. Alpha Methyldopa increases the likelihood of postnatal depression and other maternal side-effects such as sedation, postural hypotension and postnatal depression and women should therefore switch to another antihypertensive medicine if their blood pressure remains high [52–54] Clinical evaluation and diligent use of antihypertensive drugs for pre-eclampsia and eclampsia management throughout the extended continuum of care is important in reducing poor fetal and maternal outcomes.

## Limitations

Our findings need to be interpreted with caution. In certain cases, onset of pre-eclampsia may have occurred a few days before hospital referral or admission. The lag between onset of pre-eclampsia and diagnosis during referral or admission may have resulted in misclassifying some EO-PE as LO-PE at 34 weeks–particularly in the last week of EO-PE and the beginning of first week of LO-PE. We were unable to get robust data on the gestation age at ANC booking due to missing data within the in-patient records. Due to a lack of routine record-keeping (e.g. scant information on referral and clinical forms), we were also unable to collect specific data that would have helped assess more of the clinical management aspects of women referred to KNH from elsewhere, particularly prior drug dosages, administration routes, and treatment. This information would have allowed better understanding of which women received a pre-referral loading dose of $MgSO_4$, as well as other treatment or clinical management. It was also difficult to retrieve and match newborn files to those of their mothers due to the siloed filing systems. Because KNH is a national referral hospital for the most complex cases, there is a relatively high proportion of NICU infants, regardless of pre-eclampsia or eclampsia onset. On reflection, it might have been helpful to collect information on the proportions of women with EO-PE and LO-PE who came to KNH with a live baby and those who had a stillbirth before arriving, and to understand variations and effects of pre-eclampsia management across the pregnancy continuum.

## Conclusion

Early onset pre-eclampsia is associated with greater disease severity as well as adverse maternal and perinatal outcomes. Moreover, women presenting with eclampsia during the postnatal period are also at risk of poor maternal outcomes. It is critical for women and their families to

be more aware of danger signs, but also for lower level facilities to be able to recognize symptoms early and refer in a timely manner. In order to mitigate adverse outcomes, it is critical for providers to recognize the importance of early screening and history-taking, blood pressure measurement, and consistent and effective management according to international and national guidelines.

## Author Contributions

**Conceptualization:** Omondi Ogutu, Charlotte E. Warren.

**Data curation:** Charity Ndwiga, Sripad Pooja, Alfred Osoti, Charlotte E. Warren.

**Formal analysis:** Charity Ndwiga, George Odwe, Sripad Pooja.

**Funding acquisition:** Charlotte E. Warren.

**Investigation:** Omondi Ogutu, Alfred Osoti, Charlotte E. Warren.

**Methodology:** Charity Ndwiga, Omondi Ogutu, Alfred Osoti, Charlotte E. Warren.

**Project administration:** Charity Ndwiga.

**Supervision:** Alfred Osoti, Charlotte E. Warren.

**Validation:** Charity Ndwiga, Sripad Pooja, Omondi Ogutu, Charlotte E. Warren.

**Visualization:** Omondi Ogutu.

**Writing – original draft:** Charity Ndwiga.

**Writing – review & editing:** George Odwe, Sripad Pooja, Omondi Ogutu, Alfred Osoti, Charlotte E. Warren.

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
