## [Decision Letter · Decision Letter 0]

30 Jan 2020

PONE-D-20-01091

Clinical Presentation and Outcomes of Pre-Eclampsia and Eclampsia at a National Hospital, Kenya: A Retrospective Cohort Study

PLOS ONE

Dear Mrs Ndwiga,

Thank you for submitting your manuscript to PLOS ONE. After careful consideration, we feel that it has merit but does not fully meet PLOS ONE’s publication criteria as it currently stands. Therefore, we invite you to submit a revised version of the manuscript that addresses the points raised during the review process.

SPECIFIC ACADEMIC EDITOR COMMENTS: Two experts in the field reviewed your manuscript. Your study was deemed important, however major comments were raised that require the authors' attention. These comments include the need to clearly phrase the aims of this study and clarify some comments including the conclusion; there are questions about the sample size and randomization; and several of the findings need to be discussed in more detail.

We would appreciate receiving your revised manuscript by Mar 15 2020 11:59PM. To enhance the reproducibility of your results, we recommend that if applicable you deposit your laboratory protocols in protocols.io, where a protocol can be assigned its own identifier (DOI) such that it can be cited independently in the future. For instructions see: http://journals.plos.org/plosone/s/submission-guidelines#loc-laboratory-protocols

We look forward to receiving your revised manuscript.

Kind regards,

Frank T. Spradley

Academic Editor

PLOS ONE

2. Please note that all PLOS journals ask authors to adhere to our policies for sharing of data and materials: https://journals.plos.org/plosone/s/data-availability. According to PLOS ONE’s Data Availability policy, we require that the minimal dataset underlying results reported in the submission must be made immediately and freely available at the time of publication. As such, please remove any instances of 'unpublished data' or 'results not shown' in your manuscript and replace these with either the relevant data (in the form of additional figures, tables or descriptive text, as appropriate), a citation to where the data can be found, or remove altogether any statements supported by data not presented in the manuscript.

4. Your ethics statement must appear in the Methods section of your manuscript. If your ethics statement is written in any section besides the Methods, please move it to the Methods section and delete it from any other section. Please also ensure that your ethics statement is included in your manuscript, as the ethics section of your online submission will not be published alongside your manuscript.

5. Please include a caption for Figures 1, 2, 3.

Reviewers' comments:

Reviewer's Responses to Questions

**Comments to the Author**

1. Is the manuscript technically sound, and do the data support the conclusions?

Reviewer #1: Yes

Reviewer #2: Partly

2. Has the statistical analysis been performed appropriately and rigorously? 

Reviewer #1: Yes

Reviewer #2: Yes

3. Have the authors made all data underlying the findings in their manuscript fully available?

Reviewer #1: Yes

Reviewer #2: Yes

4. Is the manuscript presented in an intelligible fashion and written in standard English?

Reviewer #1: Yes

Reviewer #2: Yes

5. Review Comments to the Author

Reviewer #1: This study demonstrates an extensive exploration of the clinical outcomes of both preeclampsia and eclampsia, which are major hypertensive disorders associated with maternal and neonatal morbidity and mortality. All ethical considerations have been adhered to. Sufficient details have been provided for the study design, sampling and data analyses. Data presented is aligned to the aims of the study and highlights the seriousness of these disorders based on its onset (early vs late) during pregnancy. The statistical analyses and presentation of the data, is described sufficiently and relevant to the aim of the study.

The discussion supports the results presented and provides a landscape of both preeclampsia and eclampsia based on the largest referral hospital in Kenya.

Reviewer #2: Thank you for the opportunity to review this interesting manuscript – I commend the authors on their contribution to an extremely important yet understudied area of health research. These data add to the evidence to improve understanding of maternal morbidity and mortality due to pre-eclampsia in Sub-Saharan Africa. I have a few suggestions to enhance the study.

I think the purpose of the study could be more clearly outlined in the background. Are the authors trying to identify risk factors to stratify care or deficits in care which could be targeted for improvement? Or simply to characterise the disease burden and compare to other reported studies?

Late onset pre-eclampsia is usually defined as >37 weeks – could the authors clarify if this is what they have assessed, and add to the abstract. This would be preferable to allow comparison with other studies.

How are 100 babies admitted to NBU daily if there are only 60 beds?!

Were ISSHP criteria used to diagnose PE – please elaborate on ‘end organ damage’

Sample size – first sentence please clarify that you are trying to detect a difference between early and late. However, this seems to be an unusual approach to a sample size calculation. I would have thought that an approach to report how representative the results are of the true population would be more appropriate

Please give details about randomisation and justify why all women with eclampsia were included. I am not convinced that this will not confound your findings!

It would be helpful to give information to the uninformed reader about the funding for maternal care in Kenya – were all the women self-funding or were some state funded. Is NBU care provided for all?

The majority of women with early onset PE were multiparous – which is an important finding and should be discussed in more details.

I also note that the proportion of women with creatinine >100 were very high – especially in the late onset PE group. This could be highlighted and discussed in more detail. Were these cases AKI or CKD or unknown?

I am very surprised at the low proportion of women with Hb <10 - even in the context of 31% HELLP syndrome – this should be discussed.

Why were so many women with LO-PE delivered by CS? Were there differences in care l complications?

I am confused about the number of women with renal failure and those needing dialysis being different. Please could this be clarified.

What were the definitions of neonatal complications?

Some of the detail in the hypertension treatment section could be moved to the discussion or omitted.

The authors suggest that early referral would improve outcomes – but I am not clear how these data support that recommendation (although I am sure it is correct!). Are any information available about pathways of referral, and women from outlying districts having worse outcomes?

I am also unclear about the conclusion that women with postpartum presentation have worse outcomes – could this be highlighted more in the results?

Minor Comments

First line paragraph 2 background is repetitive and could be omitted

Was LMP used to calculate gestational age – when the methods describe ‘date’

Table 1 – Gestational age at onset – do the n= values need to be repeated here?

Figure 3 – How many women had two risk factors e.g. both previous PE and Chronic hypertension?

Table 2 – why is lowest dBP reported?

Results – pre-eclampsia characteristics – please add units. Suggest rewrite the last two sentences are currently unclear.

Table 3 should be Table 4 in the text

6. PLOS authors have the option to publish the peer review history of their article (what does this mean?). If published, this will include your full peer review and any attached files.

Reviewer #1: No

Reviewer #2: No

---

## [Author Response · Author response to Decision Letter 0]

1 Apr 2020

Thank you. 

 please find attached authors response to reviewers comments attached at "general information tab" on authors submission menu

 Authors response to reviewers' comments 3/2/2020

Reviewers' comments:

4. Your ethics statement must appear in the Methods section of your manuscript. If your ethics statement is written in any section besides the Methods, please move it to the Methods section and delete it from any other section. Please also ensure that your ethics statement is included in your manuscript, as the ethics section of your online submission will not be published alongside your manuscript.

Authors’ response: 

Thank you. The ethics statement has been moved from the abstract and take to the method section page 6 line 133- 135. 

Reviewers' comments:

5. Please include a caption for Figures 1, 2, 3.

Authors’ response: 

This has been done as follows: Figure 1; page 5 line 11-118, Figure 2; page 7, line 198-201, Figure 3; page 9, line 256-259.

Reviewers' comments:

Reviewer's Responses to Questions

Comments to the Author

Reviewers' comments:

1. Is the manuscript technically sound, and do the data support the conclusions?

Reviewer #1: Yes

Reviewer #2: Partly

Authors’ response: 

Thank you. The conclusion draws from the data presented. 

 Reviewers' comments:

2. Has the statistical analysis been performed appropriately and rigorously? 

Reviewer #1: Yes

Reviewer #2: Yes

Authors’ response: 

Thank you 

Major comments

 Reviewers' comments:

2. Have the authors made all data underlying the findings in their manuscript fully available?

The PLOS Data policy requires authors to make all data underlying the findings described in their manuscript fully available without restriction, with rare exception (please refer to the Data Availability Statement in the manuscript PDF file). The data should be provided as part of the manuscript or its supporting information or deposited to a public repository. For example, in addition to summary statistics, the data points behind means, medians and variance measures should be available. If there are restrictions on publicly sharing data—e.g. participant privacy or use of data from a third party—those must be specified.

Reviewer #1: Yes

Reviewer #2: Yes

Authors’ response: 

Thank you 

Reviewers' comments:

4. Is the manuscript presented in an intelligible fashion and written in standard English?

Reviewer #1: Yes

Reviewer #2: Yes

Authors’ response: 

Thank you 

 5. Review Comments to the Author

Reviewer #1: This study demonstrates an extensive exploration of the clinical outcomes of both preeclampsia and eclampsia, which are major hypertensive disorders associated with maternal and neonatal morbidity and mortality. All ethical considerations have been adhered to. Sufficient details have been provided for the study design, sampling and data analyses. Data presented is aligned to the aims of the study and highlights the seriousness of these disorders based on its onset (early vs late) during pregnancy. The statistical analyses and presentation of the data is described sufficiently and relevant to the aim of the study.

Authors’ response: 

Thank you for your review and positive feedback. 

Review Comments to the Author 

The discussion supports the results presented and provides a landscape of both preeclampsia and eclampsia based on the largest referral hospital in Kenya.

Authors’ response: 

Thank you for your comments

Review Comments to the Author 

Reviewer #2: Thank you for the opportunity to review this interesting manuscript – I commend the authors on their contribution to an extremely important yet understudied area of health research. These data add to the evidence to improve understanding of maternal morbidity and mortality due to pre-eclampsia in Sub-Saharan Africa. I have a few suggestions to enhance the study.

I think the purpose of the study could be more clearly outlined in the background. Are the authors trying to identify risk factors to stratify care or deficits in care which could be targeted for improvement? Or simply to characterise the disease burden and compare to other reported studies?

Authors’ response: 

Thanks for your review and feedback. Our study aims to describe the characteristics and compares the outcomes between early onset of pre-eclampsia and late onset of pre-eclampsia. We are describing the disease burden which has not been described before in this setting, to raise awareness about the two sub-types of PE and the associated outcomes that can be compared with other reported studies. Please see our study objectives page 2 line 34 and 35.

Review Comments to the Author 

Late onset pre-eclampsia is usually defined as >37 weeks – could the authors clarify if this is what they have assessed and add to the abstract. This would be preferable to allow comparison with other studies.

Authors’ response: 

Regarding the classification of PE, we used ISSHP criteria for diagnosis and classification os pre-eclampsia- early onset pre-eclampsia (EO-PE: less than 34 weeks gestation) while late onset pre-eclampsia, (LO-PE: greater than or equal to 34 weeks gestation) based on Tranquilli AL, etal 2013 and Brown et al., 2018 recommendation. We inserted the weeks in the abstract, as you have suggested, to help distinguish early versus late onset of pre-eclampsia. Please see page 2, line 39 and 40 and at the background section page 5, line 104 to 105. 

Review Comments to the Author 

How are 100 babies admitted to NBU daily if there are only 60 beds?!

Authors’ response: 

Thank you for highlighting the number of babies admitted at the NBU. We wanted to indicate that the NBU at the referral hospital is sometimes overstretched, admitting more that its carrying capacity. We have added a sentence that reads; “As the largest referral hospital in this Country, patients are never turned away and sometimes, the NBU ward admits patients more than its capacity.’ Please see page 6, line 143 -145.

Review Comments to the Author 

Were ISSHP criteria used to diagnose PE – please elaborate on ‘end organ damage’

Authors’ response: 

Thank you for highlighting this. Indeed, ISSHP criteria was used to diagnose and classify pre-eclampsia based on Brown M.A et al., 2018. Similarly, we used ISSHP criteria to diagnose end organ damage such as chest pain that may be as a result of myocardial ischemia or infarction, back pain may denote aortic dissection; and dyspnea may suggest pulmonary edema or congestive heart failure was used. In addition, a patient with these symptoms may present with neurologic symptoms such seizures, visual disturbances, and altered level of consciousness and may be indicative of hypertensive encephalopathy. We have added more explanation in the paper, please see page 6, line 159 to 162.

Review Comments to the Author 

Sample size – first sentence please clarify that you are trying to detect a difference between early and late. However, this seems to be an unusual approach to a sample size calculation. I would have thought that an approach to report how representative the results are of the true population would be more appropriate 

Authors’ response: 

Our study is based on retrospective review of the patients records, as expected, we do have some limitations considering secondary review of the patient records. We believe our sampling approach would be representative 

Review Comments to the Author 

It would be helpful to give information to the uninformed reader about the funding for maternal care in Kenya – were all the women self-funding or were some state funded.

Authors’ response: 

Thank you for your helpful suggestion. We have added texts below about funding maternal care in Kenya as follows; Maternal and newborn services KNH are free to the user since Kenya introduced free maternal and newborn, child health services in June 2013. This aims at promoting health facility delivery service utilization and reducing pregnancy-related mortality in the country (Bourbonnais, 2013 Gitobu et al., 2018). However, this fee does not cater for medical investigations beyond the recommend ANC profile hence women pay for extra cost when required. Please see page 6, line 142 to 153.

Review Comments to the Author 

Please give details about randomization and justify why all women with eclampsia were included. I am not convinced that this will not confound your findings!

Authors’ response: 

Thank you for your comment. We randomly sampled the inpatient records describing pre-eclampsia, stratified by month and year of observation. All Eclampsia cases were included because they were very few. Since we are only interested in the characteristics, we the effects confounders are likely to be minimal. 

Review Comments to the Author 

Is NBU care provided for all? 

Authors’ response: 

Not all babies receive NBU care, only babies with complications get admitted to NBU while well babies’ room-in with their mothers’ in the postnatal ward. Please see page 6, line 143 -144.

Review Comments to the Author 

The majority of women with early onset PE were multiparous – which is an important finding and should be discussed in more details. High parity may have been associated with early onset and more likely recurrent PE since we receive a high number of referred patients with prior history of preeclampsia. 

Authors’ response: 

Thank you, for this observation. 

Our study shows that multiparous women were slightly more likely to present with EOPE compared to non-multiparous women. Unlike our findings, pre-eclampsia is associated with first pregnancies (Kimberly and Nagaraj 2017; Luo etal 2007; Macdonald-Wallis etal 2011). While change in paternity is risk factor in multi parous women (Tubbergen P. etal 1999). Other studies (Muniro, Z., etal., 2019; Kashanian, M. etal 2011) history of pre-eclampsia and hypertension are risk factors to pre-eclampsia. Our results show 317 women (51%) had 2 or more births, multiparous women were more likely to experience chronic HTN (75%) and prior pre-eclampsia (91%) compared to non-multiparous women Please see discussion page 12 line, 312 -330.

Review Comments to the Author 

I also note that the proportion of women with creatinine >100 were very high – especially in the late onset PE group. This could be highlighted and discussed in more detail. Were these cases AKI or CKD or unknown? I am very surprised at the low proportion of women with Hb <10 g/dL- even in the context of 31% HELLP syndrome – this should be discussed.

Authors’ response: 

Also, the study site routinely receives critical patients with organ failure including renal dysfunction. Majority of the renal dysfunction are acute and are resolved on observation. severe forms of AKI require dialysis. However, specifics of organ dysfunction, management and subsequent outcomes is beyond the scope of this current publication. The proportion of women in our sample that had HELLP syndrome were 16% (39 women), out of these, 20 of these women had Hb<10. Our results therefore agree with what is documented in literature that women HELLP are more likely to associated with low HB. Please see page 14, line 363 to 370. 

Review Comments to the Author 

Why were so many women with LO-PE delivered by CS? Were there differences in care l complications?

Authors’ response: 

Thank you, this discussed as follows: 

We did not explore the reasons but other studies - Parmar D et al 2018 -over 40%: Akech

etal.,2012 show high prevalence of C/S in KNH due to co-morbidities and late referrals from other facilities. Please see page 12, line 341 -344. 

Review Comments to the Author 

I am confused about the number of women with renal failure and those needing dialysis being different. Please could this be clarified. 

Authors’ response: 

Thank for your comment. 

Our results show that 26 women were diagnosed with renal failure. Out of these, 14 had dialysis performed. It appears some of the renal failure cases were conservatively managed. Please see page 11, line 276 to 276 and Table 4, titled “Neonatal and maternal complications renal failure and dialysis highlighted in red line 278 to 280.

Review Comments to the Author 

What were the definitions of neonatal complications?

Authors’ response: 

Thank you for your comments 

Neonatal complications were defined as respiratory distress syndrome (RDS), neonatal jaundice, asphyxia, neonatal convulsion, encephalopathy, intraventricular hemorrhage (IVH), neonatal convulsion or sepsis and necrotizing enterocolitis (NEC). Some of these complications are not reported in this manuscript.

Reviewer comments: 

Review Comments to the Author 

Some of the detail in the hypertension treatment section could be moved to the discussion or omitted.

Thank you. We have omitted some of the hypertension treatment details. Please see page 11& 12, line 288 to 299. 

Review Comments to the Author 

The authors suggest that early referral would improve outcomes – but I am not clear how these data support that recommendation (although I am sure it is correct!). Are any information available about pathways of referral, and women from outlying districts having worse outcomes? I am also unclear about the conclusion that women with postpartum presentation have worse outcomes – could this be highlighted more in the results?

Authors’ response: 

We did no capture data on referral process; however, we have information on place of residence which was used as a proxy for referral. Women residing from other counties outside Nairobi were considered referrals. Further analysis (result not shown) shows that women from residing from other counties (126) had relatively poor maternal outcomes (deaths and PPH) compared to their counterparts in Nairobi. This may be attributed to delay in referrals as some of the cases arrived very late at KNH. Please see page 14, line 373 to 373.

Minor Comments

Review Comments to the Author 

First line paragraph 2 background is repetitive and could be omitted

Authors’ response: 

Thank you for your comment. This has been omitted. 

Review Comments to the Author 

Was LMP used to calculate gestational age – when the methods describe ‘date’

Authors’ response: 

Yes, LMP was used to calculate gestational age. We have in include this “determined by the last monthly period and ultrasound where available” to clarify see page 8 line 202.

Review Comments to the Author 

Table 1 – Gestational age at onset – do the n= values need to be repeated here?

Authors’ response: 

Thank you, this has been removed

Review Comments to the Author 

Figure 3 – How many women had two risk factors e.g. both previous PE and Chronic hypertension?

Authors’ response: 

Thank you. We have Included proportion of women who experienced both previous PE and Chronic hypertension. In Figure 3 page 9, line 256 - 259

Review Comments to the Author 

Table 2 – why is lowest dBP reported? It import

Authors’ response: 

Thank you for noting this inconsistency. 

We have reported the Highest dBP in lieu of Lowest dBP. Please see page 11, line 270 -271 in track changes and in Table 2, same page line 274- 275 highlighted in red. 

Review Comments to the Author 

Review Comments to the Author Results – pre-eclampsia characteristics – please add units. Suggest rewrite the last two sentences are currently unclear.

Authors’ response: 

We have added units to alter the sentence to read; Results show no significant differences among women with low hemoglobin level <10 g/dL between EO-PE and LO-PE line 253 and 254.

Review Comments to the Author 

Table 3 should be Table 4 in the text

Authors’ response: 

Thank you. We find that Table 3 is correctly labelled in the text. Please see Table 3 page 10, line 281- 282 and Table 4 on page 11, line 274- 298.

---

## [Decision Letter · Decision Letter 1]

4 May 2020

Clinical Presentation and Outcomes of Pre-Eclampsia and Eclampsia at a National Hospital, Kenya: A Retrospective Cohort Study

PONE-D-20-01091R1

Dear Dr. Ndwiga,

We are pleased to inform you that your manuscript has been judged scientifically suitable for publication and will be formally accepted for publication once it complies with all outstanding technical requirements.

With kind regards,

Frank T. Spradley

Academic Editor

PLOS ONE

Reviewers' comments:

Reviewer's Responses to Questions

**Comments to the Author**

1. If the authors have adequately addressed your comments raised in a previous round of review and you feel that this manuscript is now acceptable for publication, you may indicate that here to bypass the “Comments to the Author” section, enter your conflict of interest statement in the “Confidential to Editor” section, and submit your "Accept" recommendation.

Reviewer #2: All comments have been addressed

2. Is the manuscript technically sound, and do the data support the conclusions?

Reviewer #2: Yes

3. Has the statistical analysis been performed appropriately and rigorously? 

Reviewer #2: Yes

4. Have the authors made all data underlying the findings in their manuscript fully available?

Reviewer #2: Yes

5. Is the manuscript presented in an intelligible fashion and written in standard English?

Reviewer #2: Yes

6. Review Comments to the Author

Reviewer #2: No further comments to add - the authors have sufficiently addressed my concerns and the manuscript is improved.

7. PLOS authors have the option to publish the peer review history of their article (what does this mean?). If published, this will include your full peer review and any attached files.

Reviewer #2: Yes: Dr Kate Bramham

---

## [Editor Report · Acceptance letter]

11 May 2020

PONE-D-20-01091R1 

Clinical Presentation and Outcomes of Pre-Eclampsia and Eclampsia at a National Hospital, Kenya: A Retrospective Cohort Study 

Dear Dr. Ndwiga:

I am pleased to inform you that your manuscript has been deemed suitable for publication in PLOS ONE. Congratulations! Your manuscript is now with our production department. 

With kind regards,

on behalf of

Dr. Frank T. Spradley 

Academic Editor

PLOS ONE